# Characterization of Robertsonian and Reciprocal Translocations in Cattle through NGS

**DOI:** 10.3390/ani13193018

**Published:** 2023-09-26

**Authors:** Alessandra Iannuzzi, Ramona Pistucci, Angela Perucatti, Michele Zannotti, Leopoldo Iannuzzi, Pietro Parma

**Affiliations:** 1Institute for Animal Production System in Mediterranean Environment (ISPAAM), National Research Council (CNR), 80055 Portici, Italy; ramonapistucci@gmail.com (R.P.); angela.perucatti@cnr.it (A.P.); leopiannuzzi949@gmail.com (L.I.); 2Department of Agricultural and Environmental Sciences, University of Milan, 20133 Milan, Italy; michele.zannotti@unimi.it (M.Z.); pietro.parma@unimi.it (P.P.)

**Keywords:** cattle, translocation, NGS, cytogenetics, chromosome banding, FISH

## Abstract

**Simple Summary:**

This study introduces a novel approach that combines next-generation sequencing (NGS) with cytogenetic technologies to identify chromosomes involved in chromosomal anomalies in cattle. This research focuses on a chromosome anomaly discovered in male Alpine Grey cattle, as well as two previously reported cases of reciprocal translocations (rcp), namely rcp(9;11) and rcp(4;7). Traditional cytogenetic analyses, including Giemsa staining, CBA-banding, RBA-banding, and FISH techniques, were performed. Chromosomes were microdissected from conventional preparations, and the amplified products were sequenced using NGS. The sequencing reads were mapped to the reference genome, and the leverage effect was calculated to identify abnormal reads/Mb values. The results unveiled the specific chromosomes involved in the Alpine Grey cattle anomaly, which were further validated through RBA-banding and FISH analysis. Moreover, the feasibility of this approach on preserved metaphases was demonstrated by analyzing old slides from previously characterized cases.

**Abstract:**

This study presents a novel approach that combines next-generation sequencing (NGS) and cytogenetic technologies for identifying chromosomes involved in chromosomal anomalies. This research focuses on a chromosome anomaly discovered in male Alpine Grey cattle, as well as two previously reported cases of reciprocal translocations (rcps), namely rcp(9;11) and rcp(4;7). Abnormal chromosomes from Alpine Grey cattle were microdissected from conventional preparations, and the amplified products were sequenced using NGS. The sequencing reads were then mapped to the reference genome, and the leverage effect was calculated to identify abnormal reads/Mb values. The result revealed the presence of rob(26;29), which was further confirmed through traditional cytogenetic analyses such as Giemsa staining, CBA-banding, RBA-banding, and FISH techniques. Furthermore, the feasibility of this approach on preserved metaphases was demonstrated through analysis of old slides from previously characterized cases. The study highlights the challenges involved in identifying and characterizing chromosomal aberrations in bovine species and offers a potential solution for analyzing historical anomalies when fresh blood material is unavailable. The combination of NGS and cytogenetic techniques provides a cost-effective and reliable approach for characterizing chromosomal anomalies in various species, including those identified before the availability of modern banding technologies and FISH mapping using specific molecular markers.

## 1. Introduction

The field of clinical cytogenetics in cattle has made significant progress since the discovery of the Robertsonian translocation involving chromosomes 1 and 29 (rob1;29) by Gustavsson and Rockborn in 1964 [1]. The identification and characterization of chromosomal anomalies, including Robertsonian translocations (robs) and reciprocal translocations (rcps), have been the focus of extensive research due to their negative impact on reproductive efficiency in cattle. A comprehensive review by Iannuzzi et al. in 2021 [2] provides an overview of the numerous chromosomal anomalies identified in cattle and other domestic bovid species, such as river buffalo, sheep, goat, and zebu.

Initially, the challenge lay in accurately identifying the specific chromosomes involved in robs or rcps. One early method involved measuring the length of chromosomes during the metaphase and comparing them to reference measurements derived from karyotypes. For instance, the rob(1;28) was identified using this approach [3]. However, without further analysis using more precise techniques, there was no guarantee of correct chromosome identification. An improvement came with the analysis of micro-densitometry profiles of chromosomes after banding. This technique was used to initially characterize a rob anomaly in Alpine Grey cattle as rob(25;27) [4]. However, subsequent investigations employing various banding techniques and FISH technologies identified the anomaly as rob(26;29) [5]. The development of reproducible banding technologies and the establishment of the standard karyotype [6] greatly enhanced the precise identification of chromosomes involved in robs and rcps. G-banding, obtained through trypsin treatment, and particularly the R-banding obtained through late BrdU-incorporation, notably improved the banding resolution and enhanced the identification of chromosomal abnormalities in various domestic species [7,8,9,10,11,12].

However, the analysis of anomalies using FISH or/and banding techniques requires high-quality metaphases. Often, chromosomal anomalies are detected during karyological screening using Giemsa staining, and it may not be feasible to obtain new material for preparing banded chromosomes. In some cases, samples received from breeders may be of poor quality, resulting in a limited number of metaphases that cannot be adequately analyzed using FISH. Additionally, old slides of previously characterized anomalies may be available, but it is impossible to obtain suitable material for new cultures.

In this study, we introduce a pioneering approach that combines NGS with cytogenetic technologies to identify chromosomes involved in a Robertsonian translocation and two previously reported cases of reciprocal translocations: rcp(9;11) and rcp(4;7) [13,14]. This novel approach offers a solution to overcome the limitations associated with conventional methods when analyzing chromosomal anomalies in cattle.

## 2. Materials and Methods

### 2.1. Cytogenetic Analyses

A male Alpine Grey cow was included in a routine karyological screening program. Cell cultures were established following the standard methodology [15], and metaphases were stained with Giemsa. For CBA-banding patterns, normal cultures were processed according to the protocol described by Iannuzzi and Di Berardino [16] and subsequently stained with acridine orange. RBA-banded metaphases were generated as reported [16]. Briefly, cell cultures were exposed to 5-BrdU (15 µg/mL) and Hoescht33258 (30 µg/mL) for six hours before harvesting. Colcemid treatment was applied during the last hour. After hypotonic treatment and three fixations in acetic acid/methanol (1:3), cell suspensions were spread on slides. The slides were stained with acridine orange (0.1%) for 10 min, washed in tap and distilled water, and finally mounted in P-buffer with a sealed cover slip. High-resolution RBA-banding patterns were obtained and consistent results were ensured using a fluorescence microscope (Leica microscope station). The FISH procedure followed the methodology described by De Lorenzi and colleagues [13]. Two Bacterial Artificial Chromosome (BAC) probes, 312A6 (BTA26) and 953A11 (BTA29) from the INRA library [17], were used for the FISH analysis. Additionally, bovine satellites DNA SAT I and SAT IV were utilized as previously described in the context of the rob(26;29) chromosome aberration [18].

### 2.2. Microdissection

Two drops of conventional chromosome preparations were spread on coverslips, stained with Giemsa, and air-dried. The coverslips were then placed on a special support for the inverted microscope (Leica-Germany). Metaphase preparations were initially located using a 10× lens and subsequently with a 100× lens to aid in chromosome dissection. A micromanipulator and a micro-needle with a 1 µ tip were employed for precise chromosome dissection. For each sample, eight microdissected chromosomes were carefully collected and resuspended in 8 µL of TE 10:1 solution. Special care was taken to exclusively microdissect the rob chromosome derived from the translocation. This same microdissection procedure was also applied to old slides from carriers of rcp(4;7) and rcp(9;11) [13,14], where der7 and der9 were microdissected.

### 2.3. Genome Amplification

The micro-dissected chromosomes were subjected to genome amplification using the WGA4 kit (Sigma-Aldrich, Darmstadt, Germany) following the manufacturer’s instructions. The resulting product underwent purification using the GenElute™ PCR Clean-Up Kit (Sigma-Aldrich). Subsequently, a final amplification step was performed using WGA3 (Sigma-Aldrich), again following the manufacturer’s instructions. The final product was purified once more using the GenElute™ PCR Clean-Up Kit (Sigma-Aldrich) before proceeding to the sequencing step. This amplification procedure was applied to all samples.

### 2.4. NGS

DNA libraries (350 bp) were constructed for Illumina/BGI sequencing for each accession following to the manufacturer’s specifications. Subsequently, sequencing was carried out on an Illumina HiSeq XTen/NovaSeq/BGI platform through a commercial service provider (Biomarker Technologies, Beijing, China), with 150 bp read lengths. Following sequencing, the reads were mapped to the reference genome (UMD_3.1.1) using the bwa-mem2 software (2.2 version) [19]. The raw data (raw reads) of fastq format underwent initial processing using fastp software (0.21.0 version). During this step, clean data (clean reads) were generated by removing reads containing adapters, reads containing ploy-N, and low-quality reads from the raw data. Furthermore, any reads that did not map on the bovine genome were eliminated in this processing step. Subsequent downstream analyses were conducted using the clean, high-quality data. 

### 2.5. Identification of Chromosomes Involved

To identify the chromosomes involved in the chromosomal anomalies, we calculated the leverage effect (Le) of the reads/Mb value using the following formula: $$Le=\left(\frac{1}{n}+\frac{1}{\left(n-1\right)}\times {\left(\frac{x-avg}{sd}\right)}^{2}\right)\times n$$
where:*n*: the number of chromosomes (30:29 autosomes plus BTAX).*x*: the reads/Mb mapped on a specific chromosome.*avg*: the average value of the reads/Mb.*sd*: the standard deviation of the reads/Mb.

It is worth noting that the average value of the reads/Mb calculated using the formula is equal to 2. Therefore, any values greater than 4 are considered anomalous and indicative of potential chromosomal involvement in the identified anomalies. This approach represents a mathematical derivation of previously reported methods [20]. It is a widely used procedure across various fields to validate collected data and exclude any outliers. To the best of our knowledge, this is the first instance of applying this procedure to genomic data.

### 2.6. Minimum Identifiable Size of the Genomic Fragment Involved

We used a simple correlation analysis to determine the suggested approach’s detection limit for genomic regions associated with anomalies. We recognize that experimental quality, defined as the ratio of average reads for the chromosomes involved to overall average reads, is critical. Low quality can enhance background noise, thereby masking any signals from genetic peaks or fragments. With specific calculations (supplied in the Appendix A), the simulation of the number of reads necessary to obtain an Le value of 4 forms the basis of this strategy.

For more information, please refer to the Appendix A.

## 3. Results

The observation of Giemsa-stained metaphases obtained from Alpine Grey cattle revealed a karyotype of 2*n* = 59, XY, characterized by the presence of an abnormal small metacentric chromosome (Figure 1). 

NGS produced varying read counts, ranging from 13,456,832 to 45,793,542 reads. However, due to inevitable contaminations resulting from the microdissection procedure and genomic amplification, a portion of these reads mapped to the human and/or bacterial genome. In the three experiments conducted, the percentage of reads not derived from the microdissected products ranged from 21.64% to 80.58%. The relevant data are summarized in Table 1. Consequently, the usable reads for analysis were 8,389,821 for the Alpine Grey Cattle anomaly, 36,899,474 for rcp(4;7), and 8,326,706 for rcp(9;11).

The methodology employed to identify the chromosomes responsible for anomalies makes the assumption that, without microdissection, the sequencing reads would solely result from contamination, resulting in a constant number of reads/Mb across all chromosomes. Abnormal numbers of reads/Mb were identified by calculating the leverage effect (Table 1 and Figure 2A–C).

The results indicated the involvement of BTA26 and BTA29 in the Alpine Grey cattle anomaly, while confirming the involvement of BTA4, BTA7, BTA9, and BTA11 in rcp(4;7) and rcp(9;11), respectively (Table 1 and Figure 2B,C). These findings were further corroborated through RBA-banding (Figure 3A) and FISH analysis (Figure 3B) for rob(26;29). Additionally, the dicentric nature of the anomaly was verified through CBA-banding (Figure 3C). The HC-block found in the translocated chromosome was notably larger, at least double in size, compared to all other autosomes. This observation was further validated using bovine SAT I and SAT IV probes, which indicated FITC-signals in both chromosome arms of the translocated chromosome (Figure 3D,E, respectively). 

## 4. Discussion

Identifying and characterizing chromosomal aberrations in bovine species are complex tasks, with challenges involving both Robertsonian translocations (robs) and reciprocal translocation (rcp). While Giemsa staining allows for the detection of potential rob cases, it falls short in identifying about 16% of rcp cases [21]. Consequently, identifying the chromosomes involved in robs can be difficult and recognizing rcps that escape detection adds further complexity. While some proposals have been made to address this issue [22,23,24], a definitive solution for the latter problem remains elusive.

Traditionally, FISH and banding technologies have been central in identifying chromosomes involved in chromosomal anomalies [25]. However, both techniques have required high-quality chromosomal preparations, which are only feasible with fresh blood material. In this study, we demonstrated a novel approach that enables the analysis of old anomalies for which fresh blood is no longer available, making it impossible to create new cultures. Instead, we successfully utilized Giemsa-stained metaphase slides stored at room temperature for several years. This approach allows for the characterization of anomalies that were identified before the availability of modern banding technologies and FISH mapping using specific molecular markers, such as the rob(1;28) discovered in the early stages of modern animal cytogenetics [3].

Rob(26;29) was originally discovered many years ago in the Alpine Grey cattle through studies conducted by Giovanni et al. and later confirmed by Meo et al. [4,5]. Notably, continuous early screening efforts have been ongoing since then, and remarkably, no carrier of this anomaly has been identified since 2006. It is intriguing that despite the passage of many generations, the anomaly has persisted as a dicentric structure. The NGS analysis of the reads’ distributions obtained from this study provides valuable insights into the chromosomal involvement in the anomaly. The results clearly demonstrate the presence of the BTA26 and BTA29 in the derivative chromosome (Figure 2). These findings were further substantiated through RBA- and CBA-banding and FISH-techniques with specific molecular markers (Figure 3A–E). 

However, despite the initial genetic material belonging exclusively to BTA26 and BTA29, the microdissection procedure introduces certain challenges during sequencing. The results revealed reads that aligned with other chromosomes, originating from potential contaminations amplified through the two successive rounds of whole-genome amplification. Additionally, the presence of repeated regions further contributed to ambiguous results. While identifying the chromosomes involved in a rob is relatively straightforward, as a rob arises from the fusion of two single chromosomes, the same cannot be assumed for rcps. In cases where only a small part of a chromosome is involved in the derivative chromosome, the number of reads produced might be confused with background contaminations, leading to potential inaccuracies in the identification process.

For the two rcps analyzed (rcp(4;7) and rcp(9;11)), the microdissected derivative chromosomes contain almost all of the chromosomes involved. However, when only a minor part of a chromosome is present in the derivative chromosome, establishing a minimum threshold becomes important to ensure accurate identification.

The simulation analyses provided valuable insight into this aspect, highlighting the significance of experiment quality in determining the ability to identify smaller chromosomal fragments. Higher experiment quality, measured as the ratio between reads/Mb mapping on the involved chromosomes and the background average value, allows for the identification of smaller fragments (Table 2).

For instance, in the analysis of rcp(4;7), where the reads mapped on the involved chromosomes were 40 to 60 times more abundant than background noise, fragments as small as 15–20 Mb could be identified (Figure 4).

## 5. Conclusions

In conclusion, the combination of NGS with traditional cytogenetic techniques proved to be a valuable and innovative approach for identifying chromosomal anomalies in cattle (and other domestic animals). The ability to analyze old anomalies with limited available fresh blood material opens new avenues for the understanding and conservation of bovine species. As technology advances and the quality of experiments improves, further refinements in identifying smaller chromosomal fragments can be expected, contributing to a more comprehensive characterization of chromosomal aberrations in bovine genetics. These findings not only advance our understanding of chromosomal anomalies in cattle but also have practical implications for the cattle breeding industry. Early detection and characterization of chromosomal aberrations are crucial for informed breeding practices and maintaining the health and genetic diversity of cattle populations. As research continues to evolve, the presented approach may serve as a blueprint for similar studies in other animal species, providing valuable insights into the genetics of various agricultural and ecological systems.

Moreover, the analyses performed indicate that reliable results can be obtained even with a low number (<15,000,000) of reads and therefore a significant reduction in sequencing costs. The cost associated with this procedure is relatively modest, estimated at approximately EUR 400, in addition to the expenses related to personnel primarily engaged in microdissection.

It is worth noting that the proposed methodology is applicable when the rearranged chromosome is identifiable. This is typically the case with Robertsonian translocations. However, it may not be as straightforward for reciprocal translocations or other aberrations, such as deletions or duplications, where chromosomal identification can be more challenging. Finally, in the case of rcps, the analysis of the distribution of the reads on the analyzed derivatives can provide interesting indications on the position of the breakpoints (Appendix A).

## Figures and Tables

**Figure 1 animals-13-03018-f001:**
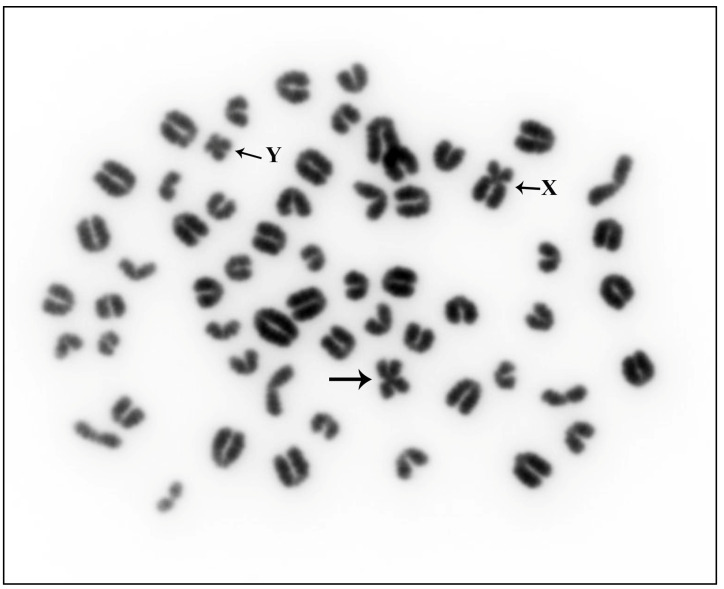
Giemsa-stained metaphase of Alpine Grey breed carrying rob(26;29) (large arrow). Sex chromosomes are also indicated.

**Figure 2 animals-13-03018-f002:**
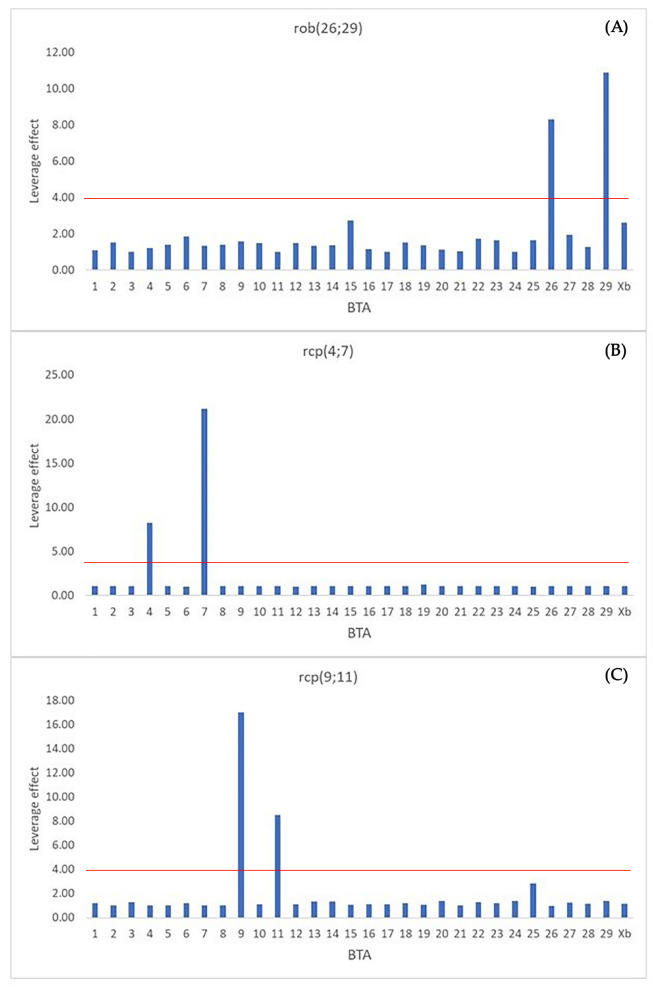
Display of leverage values in (**A**) rob(26;29), (**B**) rcp(4;7), and (**C**) rcp(9;11). The red line indicates the leverage value equal to 4.

**Figure 3 animals-13-03018-f003:**
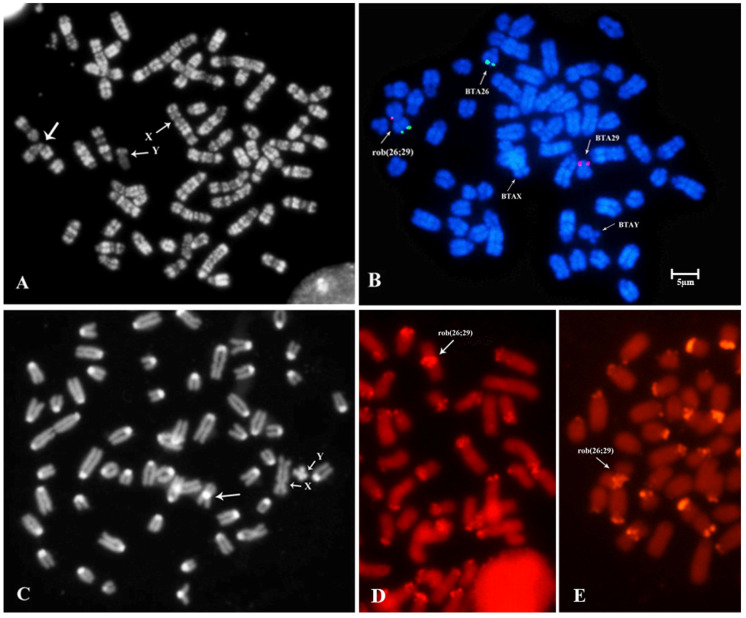
Confirmation of rob(26;29) translocation through multiple methods: (**A**) RBA-banding (large arrow); (**B**) FISH-mapping using specific molecular markers of BTA26 and BTA29; (**C**) C-banding present with a substantial block in both chromosome arms of rob(26;29), confirming its di-centric nature (large arrow); (**D**) SAT I DNA-probes showing FITC-signals in both chromosome arms of the translocated chromosome (arrow), as seen in figure; and (**E**) SAT IV DNA-probes showing FITC-signals in both chromosome arms of the translocated chromosome (arrow). Sex chromosomes are also indicated in (**A**–**C**).

**Figure 4 animals-13-03018-f004:**
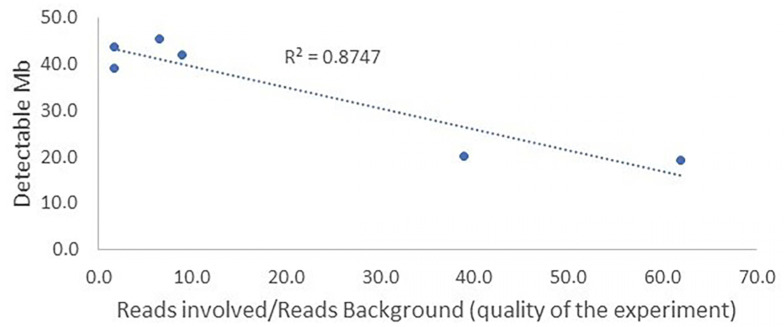
Relationship between the quality of the experiment and the minimum size of a chromosomal fragment involved in an rcp.

**Table 1 animals-13-03018-t001:** Analysis of the distribution of the reads obtained using NGS. Leverage values greater than 4 are indicated in bold.

		Reads								
Total Reads Produced	38,763,414	45,793,542	13,456,832							
% Reads Mapped on Bt Genome	0.2164	0.8058	0.6188							
Reads Mapped on BT Genome	8,389,821	36,899,474	8,326,706	Reads/Mb			Leverage Effect			
Chr	Length (bp) ^a^	rob(26;29)	rcp(4;7)	rcp(9;11)	rob(26;29)	rcp(4;7)	rcp(9;11)	rob(26;29)	rcp(4;7)	rcp(9;11)
1	158,337,067	471,590	139,222	178,910	2978	879	1130	1.08	1.10	1.23
2	137,060,424	369,232	109,916	500,154	2694	802	3649	1.50	1.10	1.04
3	121,430,405	392,202	112,301	99,084	3230	925	816	1.01	1.09	1.31
4	120,829,699	344,224	12,598,073	294,278	2849	104,263	2435	1.23	8.28	1.02
5	121,191,424	332,807	193,509	267,536	2746	1597	2208	1.40	1.08	1.04
6	119,458,736	305,613	515,666	542,937	2558	4317	4545	1.84	1.04	1.19
7	112,638,659	312,265	18,678,647	293,552	2772	165,828	2606	1.35	21.16	1.01
8	113,384,836	405,973	155,607	272,161	3580	1372	2400	1.40	1.09	1.02
9	105,708,250	387,862	137,213	1,882,849	3669	1298	17,812	1.59	1.09	17.03
10	104,305,016	377,332	124,208	168,705	3618	1191	1617	1.48	1.09	1.12
11	107,310,763	346,821	113,704	1,406,506	3232	1060	13,107	1.01	1.09	8.51
12	91,163,125	246,263	450,367	146,139	2701	4940	1603	1.49	1.04	1.12
13	84,240,350	234,151	64,854	61,383	2780	770	729	1.34	1.10	1.34
14	84,648,390	234,377	68,151	62,019	2769	805	733	1.36	1.10	1.34
15	85,296,676	343,538	84,389	156,286	4028	989	1832	2.72	1.09	1.08
16	81,724,687	237,750	129,944	349,240	2909	1590	4273	1.15	1.08	1.14
17	75,158,596	236,964	159,413	130,379	3153	2121	1735	1.00	1.07	1.10
18	66,004,023	177,235	47,761	75,037	2685	724	1137	1.52	1.10	1.23
19	64,057,457	228,237	1,914,237	128,350	3563	29,883	2004	1.37	1.29	1.06
20	72,042,655	244,959	73,482	44,628	3400	1020	619	1.13	1.09	1.38
21	71,599,096	217,968	71,810	178,520	3044	1003	2493	1.03	1.09	1.01
22	61,435,874	159,617	50,091	53,823	2598	815	876	1.73	1.10	1.30
23	52,530,062	138,444	101,262	60,085	2636	1928	1144	1.64	1.08	1.22
24	62,714,930	196,671	52,824	35,355	3136	842	564	1.00	1.10	1.40
25	42,904,170	158,400	363,047	340,371	3692	8462	7933	1.64	1.01	2.82
26	51,681,464	255,759	43,697	138,627	4949	846	2682	8.32	1.10	1.00
27	45,407,902	114,667	100,662	47,835	2525	2217	1053	1.93	1.07	1.25
28	46,312,546	130,327	54,170	65,806	2814	1170	1421	1.28	1.09	1.16
29	51,505,224	269,771	38,512	29,186	5238	748	567	10.88	1.10	1.39
X ^b^	148,823,899	345,868	101,823	211,310	2324	684	1420	2.61	1.10	1.16

^a^: The chromosome lengths were obtained from the genome description available at (https://www.ncbi.nlm.nih.gov/datasets/genome/GCF_000003055.6/, accessed on 25 November 2014). ^b^: Additionally, the number of reads obtained was divided by 1.5, as described in the Appendix A).

**Table 2 animals-13-03018-t002:** Data used for the simulation of the minimum identifiable length.

Anomaly ^1^	BTA ^2^	Reads Gen. (avg) ^3^ Reads Inv. (avg) ^4^	Ratio ^5^	Threshold Reads ^6^	Threshold Mb ^7^	
rob(26;29)	26	3024	4949	1.6	217,000	43.8
	29	3024	5238	1.7	205,000	39.1
rcp(4;7)	4	2678	104,263	38.9	2,100,000	20.1
	7	2678	165,828	61.9	3,200,000	19.3
rcp(9;11)	9	2008	17,811	8.9	750,000	42.1
	11	2008	13,106	6.5	595,000	45.4

^1^: anomaly examined. ^2^: chromosomes involved. ^3^: average of the reads/Mb value on the chromosomes not involved in the anomaly. ^4^: average of the reads/Mb value on the chromosomes involved in the anomaly. ^5^: ratio between 3 and 4 (reads inv./reads gen.). ^6^: minimum number of total reads to reach a value of Le = 4 (threshold of significance of a commonly accepted anomalous value). ^7^: ratio between ^6^ and ^4^ (threshold reads/reads inv.). This represents the minimum identifiable size of a genomic fragment possibly involved in an rcp.

## Data Availability

The datasets generated during and/or analysed during the current study are available from the corresponding author, AI, upon reasonable request.

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
