# Peer review of "Characterization of Robertsonian and Reciprocal Translocations in Cattle through NGS"

_animals, 2023, doi:10.3390/ani13193018_

Round 1
Reviewer 1 Report
Dear authors,
I found your present communication interesting and well done, presentation of results and figures are of good quality and the general shape of your article is matching quality standards for publication in a Journal as Animals.

Dear authors,
An english review of your MS might be beneficial for polishing your writing.
Author Response
Reviewer 1
Line 165. It say: “Sex chromosomes are also indicated in all three figures. Notice that the sex chromosomes are not indicated in figure B.
We have changed the figure by adding the missing indications.

Reviewer 2 Report
The manuscript by Iannuzzi et al demonstrates the power of microdissected chromosome sequencing in characterisation of translocation using cattle as a model. Authors develop a model allowing for prioritisation of chromosomes involved in translocation, validate it with simulations and test on real samples. The prediction results are validated with cytogenetics techniques. Overall, the manuscript presents a compelling story and would be suitable for publication after some technical additions and edits. Throughout the review, I will be referring to potential additional analyses towards identification of specific chromosome regions involved in translocations - while this work would be an interesting addition to the paper, I will totally understand if authors consider this work is out of scope for current version of the manuscript.
Please see more specific comments below.
Methods:
- exact sequencing platform and read lengths are not specified
- bwa algorithm and software version are not specified
- how the number of reads mapped was calculated? Was there any filtering for mapping quality etc?
- no explanations are given for the statistical interpretation of the proposed Le metric. Why average converges to 2? Why 4 is considered a significant anomaly?
Results & discussion:
lines 147-149 - it seems natural to test leverage effect values on some public cattle WGS data - some chromosomes might have lower read mapping rates due to increased proportion of repeats (e.g., chrX), and normalising for this effect could improve scoring a bit
Figure 2 - Le value for chr25 in rcp(9;11) seems a bit increased compared to other chromosomes - this could indicate a small region translocation in addition to the chr9 and chr11. The putative region could be visualised by plotting read counts per window (e.g., 1Mbp) or individual read mappings in a genome browser.
Justification of dicentric rob(26;29) is unclear to me - C-block is more likely to be mostly transposon-rich pericentric heterochromatin - this can be translocated with or without a centromere. I would expect satDNA (https://academic.oup.com/gbe/article/11/4/1152/5393269) mapping to prove that two centromeric arrays indeed exist in this case.
ll204-207 - I'm not following this: rcp(4;7) and rcp(9;11) have higher Le values compared to rob(26;29), but you state that only reciprocal translocations contain material from other chromosomes. This might make sense from biological point of view, but sequencing data shows a combination of contamination with whole-genome material and real signals - as you've indicated in the previous paragraph. Disentangling those two might be possible when accounting for read mapping density along the chromosomes - the expectation for the contamination is even but low coverage, while true translocations would be seen as short regions of increased read mapping density.
Figure 4 - information on simulation analyses should be added to methods and possibly to results (see also last point for methods section)
Author Response
Reviewer 2
Methods:
- exact sequencing platform and read lengths are not specified
We have modified this part which is now:
DNA libraries(350bp) for Illumina/BGI sequencing were constructed for each accession according to the manufacturer's specifications. After DNA library construction, sequencing was performed on an Illumina HiSeq XTen/NovaSeq/BGI platform by a commercial service (Biomarker Technologies, Beijing, China), with 150-bp read lengths. After sequencing, the reads were mapped to the reference genome (UMD_3.1.1) using the bwa-mem2 (2.2 version) software [18]. Raw data (raw reads) of fastq format were firstly processed through fastp (0.21.0 version) software. In this step, clean data (clean reads) were obtained by removing reads containing adapter, reads containing ploy-N and low-quality reads from raw data. Furthermore, even the reds that do not map on the bovine genome were eliminated in this step. All the downstream analyses were based on clean data with high quality.
- bwa algorithm and software version are not specified
We have added the version (2.2)
- how the number of reads mapped was calculated? Was there any filtering for mapping quality etc?
The number of reads is calculated directly in the analysis phase and is a data provided following the analyses. It is calculated with fastp. Also, yes, there is a filtering phase (now described in the new version)
- no explanations are given for the statistical interpretation of the proposed Le metric. Why average converges to 2? Why 4 is considered a significant anomaly?
The calculation of the le factor is a mathematical derivation of what is contained in the following article: Chatterjee, A., Hadi, A.S. Influential Observations, High Leverage Points, and Outliers in Linear Regression. Statist. Sci. 1986, 1, 379 – 393. https://doi.org/10.1214/ss/1177013622. We have added this reference.
The fact that the mean of these values is 2 is a mathematical consequence of the formula and normally the double values are considered anomalous. It is a statistical custom.
Results & discussion:
lines 147-149 - it seems natural to test leverage effect values on some public cattle WGS data - some chromosomes might have lower read mapping rates due to increased proportion of repeats (e.g., chrX), and normalising for this effect could improve scoring a bit.
Following this suggestion, we calculated the Le value on WGS experiments. In the first case, the following distribution was obtained:
In fact, chromosome X has an average of 340,000 reads/Mb against a general average of 233,000.
The same result is obtained with another dataset.
Considering what we obtained we introduced a factor of *0.67 for the number of reads obtained regarding BTAX and we thank the reviewer for this important suggestion! We have also added this information in supplementary materials.
Figure 2 - Le value for chr25 in rcp(9;11) seems a bit increased compared to other chromosomes - this could indicate a small region translocation in addition to the chr9 and chr11. The putative region could be visualised by plotting read counts per window (e.g., 1Mbp) or individual read mappings in a genome browser.
Your observation is correct. However, being the value <4 (threshold of significance considered) it is not taken into consideration. Unfortunately, a limitation of this approach is the detection limit of any small genomic fragment involved in the anomaly. This dimension is strongly influenced by the quality of the experiment, as demonstrated in the reported simulation. In the case of rcp9;11 the quality of the experiment is around 9, and therefore it is assumed that a fragment of at least 30/35 Mb can be identified. We know from the characterization of this previously published rcp that this is not possible.
However, to go into detail into this aspect we analyzed the frequency of 340,371 reads mapped to chromosome 25 (in the case of rcp9;11) with 1 Mb windows. The result showed a uniform distribution of the reds.
Justification of dicentric rob(26;29) is unclear to me - C-block is more likely to be mostly transposon-rich pericentric heterochromatin - this can be translocated with or without a centromere. I would expect satDNA (https://academic.oup.com/gbe/article/11/4/1152/5393269) mapping to prove that two centromeric arrays indeed exist in this case.
The HC-block (C-band) present at the centromeric region of the translocated chromosome is at least double in size than those of all remaining autosomes. This agrees for a dicentric translocation. However, as the reviewer suggests, we performed a FISH with two probes available in our lab containing bovine SAT I and SAT IV DNA. Clear double signals were detected in both chromosome arms of the translocated chromosome (see Figure 3D-E), further confirming the dicentric nature of rob(26;29).
For your convenience we attach here another C-banded metaphase of cattle carrying rob(26;29) where the two HC-blocks appeared separated in the translocated chromosome.
ll204-207 - I'm not following this: rcp(4;7) and rcp(9;11) have higher Le values compared to rob(26;29), but you state that only reciprocal translocations contain material from other chromosomes. This might make sense from biological point of view, but sequencing data shows a combination of contamination with whole-genome material and real signals - as you've indicated in the previous paragraph. Disentangling those two might be possible when accounting for read mapping density along the chromosomes - the expectation for the contamination is even but low coverage, while true translocations would be seen as short regions of increased read mapping density.
In reality we have stated what was reported (“only reciprocal translocations contain material from other chromosomes”) in reference to the quality of the experiment which turned out to be very low, but sufficient, in the case of rob26;29. Using this approach, the higher the quality of the experiment, the better the definition of the chromosomes, or fragments of them, involved. In a hypothetical experiment without contamination and without repetitive regions no reds are expected on uninvolved chromosomes.
However, we analyzed the distribution of the reads on some involved chromosomes and noted that this analysis can provide interesting observations on the position of the break points. This result has been added to the text and better specified in the supplementary results.
Figure 4 - information on simulation analyses should be added to methods and possibly to results (see also last point for methods section)
We have added a paragraph in materials and methods and new data and better explanation.

Reviewer 3 Report
The paper introduces a novel approach that combines next-generation sequencing (NGS) and cytogenetics to identify chromosomal anomalies in cattle. While this approach shows potential, several important issues must be addressed to enhance its validation and effectiveness.
1. A significant omission in the paper is the absence of direct fluorescence in situ hybridization (FISH) of microdissected DNA probes to metaphase spreads. This omission weakens the method's robustness and validation. Performing FISH experiments (with or without Cot1) could visualize the translocated chromosomes, aiding subsequent identification with Bacterial Artificial Chromosomes (BACs) and potentially rendering NGS unnecessary.
2. Repetitive presentation of the same numerical values (such as total reads produced, % and number of reads mapped on the cattle genome, leverage values) across the text, Table 1, and Figure 2 raises concerns about data clarity and presentation.
3. Another notable concern is the excessive reliance on self-citations, with 14 out of 23 references coming from the authors themselves. While self-citations can be relevant, overusing them may introduce bias and compromise the objective presentation of the research landscape.
4. To enhance the study's value,I would recommend that the authors include a quantitative comparison between the estimated costs and time requirements of their method and the combinative BAC mapping approach. This comparison would provide readers with a clearer understanding of the strengths and weaknesses of each method.
Author Response
Reviewer 3
- A significant omission in the paper is the absence of direct fluorescence in situ hybridization (FISH) of microdissected DNA probes to metaphase spreads. This omission weakens the method's robustness and validation. Performing FISH experiments (with or without Cot1) could visualize the translocated chromosomes, aiding subsequent identification with Bacterial Artificial Chromosomes (BACs) and potentially rendering NGS unnecessary.
Unfortunately, we partially agree with this observation. The suggested approach is valid when it is possible to recognize the abnormal chromosome. In the three cases that we present it was actually possible to perfectly identify the chromosomes involved without applying the proposed technologies. We have used these three cases to test a different approach, useful when the chromosome preparations are sufficient to identify the abnormal chromosome, but not to characterize it. Often in previous publications we have identified, or confirmed, the involved chromosomes by FISH with BAC probe. However, in this case a first rough classification was possible, based on the size and banding of the chromosomes involved. However, if we imagine that it is not possible to make this first classification, the fact of marking the microdissection and hybridizing it would not provide any additional information, other than confirming that we have microdissected the correct chromosome(s). This analysis was done in the original characterization of rcp4;7, as shown in the attached photo.
For example, we believe that a new analysis, with the approach we propose, would be useful regarding the recently published rob13;23: Dzitsiuk and Tipilo. CHROMOSOMAL ANOMALIES IN DAIRY CATTLE
AS REASONS OF IMPAIRED FERTILITY. Agricultural Science and Practice, 2019;6; 60-66).
- Repetitive presentation of the same numerical values (such as total reads produced, % and number of reads mapped on the cattle genome, leverage values) across the text, Table 1, and Figure 2 raises concerns about data clarity and presentation.
We agree with the reviewer that table 1 is quite complex, but we believe that the figure relating to the leverage effect values alone is not sufficient to better understand the results obtained. For this reason, if the editor agrees, we would prefer to keep table 1 in the text.
- Another notable concern is the excessive reliance on self-citations, with 14 out of 23 references coming from the authors themselves. While self-citations can be relevant, overusing them may introduce bias and compromise the objective presentation of the research landscape.
It’s correct, but joining the papers published during the last 20 years by the two groups involved in this study we reach a very high of percentage of all papers describing cattle chromosome abnormalities. In addition, the three chromosome abnormalities we studied in the present report have been found and described earlier by the two groups. Thus the numerous our self-citations can be explained.
- To enhance the study's value,I would recommend that the authors include a quantitative comparison between the estimated costs and time requirements of their method and the combinative BAC mapping approach. This comparison would provide readers with a clearer understanding of the strengths and weaknesses of each method.
We have added a short paragraph about this aspect. However, we want to underline that this procedure does not want to replace the other approaches (banding, FISH with BAC, etc) but wants to be a solution when the other methodologies are not applicable, in this sense the cost takes a back seat.

Reviewer 4 Report
In the manuscript a novel approach focused on characterization rearranged chromosomes due to Robertsonian or reciprocal translocations is presented. The main aim of the study was development of a novel approach (chromosome microdissection + next generation sequencing, NGS) for the characterization of the rearranged chromosomes. Three abnormalities were analyzed: a case of centric fusion and two cases of reciprocal translocation, previously reported. In all these cases the rearranged chromosomes had a distinguishing morphology (an additional bi-armed chromosome in a carrier of centric fusion and the longest one-armed chromosome in two carriers of reciprocal translocation). A case of a centric fusion, rob (26;29), was perfectly documented by banding and FISH techniques.
The presented idea (microdissection + NGS) is original and interesting, however, description of the applied procedures are not sufficiently clear. The following issues should be explained in the manuscript:
1. Line 101 – there is a sentence “For each sample, eight microdissected chromosomes were carefully collected ….”. Was the rearranged chromosome only scraped from each slide or surrounding chromosomes (or their fragments) were also scrapped?
2. Line 120 – was the formula of “leverage effect” developed by the authors? The formula and its usefulness in genomic studies should be explained. Are there any references regarding the use of this formula in genomic studies?
3. Line 123 – description of “n” is not clear. What does mean “observations”? What values can be achieved by “n”?
4. Lines 128-129. It should be explained why value “2” is expected for normal chromosomes, while values “>4” indicate chromosome rearrangement.
5. Lines 139-140. How it is was found that some reads were mapped to human or bacterial genomes?
6. Table 1:
a) Second column – does the length of chromosomes was taken from the reference genome annotation? If yes, it should be indicated.
b) Do the reads for all chromosomes mean that the pooled DNA of the scrapped rearranged chromosome was contaminated by all chromosomes?
7. Lines 147-149. This sentence is not clear.
8. Conclusions. It should be mentioned that the proposed approach can be applied if the rearranged chromosome can be identified, due to its morphology or size, on Giemsa stained slides.
Quality of English language is satisfactory, however, I am not a native speaker.
Author Response
Reviewer 4
- Line 101 – there is a sentence “For each sample, eight microdissected chromosomes were carefully collected ….”. Was the rearranged chromosome only scraped from each slide or surrounding chromosomes (or their fragments) were also scrapped?
We added the following sentence: Special care was taken to microdissect only the rob chromosome originating from the translocation.
- Line 120 – was the formula of “leverage effect” developed by the authors? The formula and its usefulness in genomic studies should be explained. Are there any references regarding the use of this formula in genomic studies?
The calculation of the le factor is a mathematical derivation of what is contained in the following article: Chatterjee, A., Hadi, A.S. Influential Observations, High Leverage Points, and Outliers in Linear Regression. Statist. Sci. 1986, 1, 379 – 393. https://doi.org/10.1214/ss/1177013622. We have added this reference. This procedure is used in any field to verify the data collected and to exclude any outliers. As far as we know, this is the first time that this procedure has been applied to genomic data. We have added a small sentence.
- Line 123 – description of “n” is not clear. What does mean “observations”? What values can be achieved by “n”?
We changed to: n: the number of chromosomes (30: 29 autosomes plus BTAX).
- Lines 128-129. It should be explained why value “2” is expected for normal chromosomes, while values “>4” indicate chromosome rearrangement.
In this case we cannot say normal or abnormal chromosome. It is an analysis of the frequency of the reads that map to a certain chromosome. If we imagine not carrying out any microdissection action, what we expect is a uniform distribution of reads on the genome, as result of contamination and/or repetitive sequences. What is evaluated is the presence of outliers, indicating that some parts of the genome are overrepresented (derived from microdissection). The fact that the mean of these values is 2 is a mathematical consequence of the formula and normally the double values are considered anomalous. It is a statistical custom.
- Lines 139-140. How it is was found that some reads were mapped to human or bacterial genomes?
We have added: Furthermore, even the reds that do not map on the bovine genome were eliminated in this step.
- Table 1:
- a) Second column – does the length of chromosomes was taken from the reference genome annotation? If yes, it should be indicated.
We have added this information in the legend of table 1.
- b) Do the reads for all chromosomes mean that the pooled DNA of the scrapped rearranged chromosome was contaminated by all chromosomes?
Contaminations during the microdissection step are inevitable, as is during the genomic amplification step. It cannot be excluded that some genomic fragments are collected during this phase. However, we think that most of the illegitimate mappings on the bovine genome are due to repetitive sequences.
- Lines 147-149. This sentence is not clear.
The new paragraph is: “The method used to determine which chromosomes are responsible for anomalies makes the assumption that, in the absence of microdissection, the sequencing reads would only be the product of contamination, resulting in a constant number of reads/Mb across all chromosomes.”
……Imagine starting a microdissection procedure but not microdissecting anything specific (at random).. . What is expected is only contamination or random and repetitive sequences, not a greater concentration in a genomic region.
- Conclusions. It should be mentioned that the proposed approach can be applied if the rearranged chromosome can be identified, due to its morphology or size, on Giemsa stained slides.
We added the following sentence: “Finally, the proposed methodology can be used when the rearranged chromosome is identifiable, which is always the case when there is a Robertsonian translocation but not necessarily the case when there is a reciprocal translocation or another aberration (such as a deletion or duplication).”

Round 2
Reviewer 4 Report
The manuscript was revised accordingly and the responses to all questions given in review of an initial submission are satisfactory.
There is a single issue, which should be clarified.
Line 134 (revised version) – description of “n” is still not clear. Does the the following description is correct?
n: number of chromosomes (1, 2, ... 30), to which the reads were mapped
Other comments:
line 141: should be: method (instead of methods)?
line 143: it suggested to replace "instance" by "attempt"
line 287. should be: approximately